# Migraine Aura—Catch Me If You Can with EEG and MRI—A Narrative Review

**DOI:** 10.3390/diagnostics13172844

**Published:** 2023-09-02

**Authors:** Franz Riederer, Johannes Beiersdorf, Adrian Scutelnic, Christoph J. Schankin

**Affiliations:** 1Department of Neurology, Inselspital, Bern University Hospital, University of Bern, CH 3010 Bern, Switzerlandchristoph.schankin@insel.ch (C.J.S.); 2Department of Neurology, University Hospital Zurich, Medical Faculty, University of Zurich, CH 8091 Zurich, Switzerland; 3Karl Landsteiner Institute for Clinical Epilepsy Reserach and Cognitive Neurology, AT 1130 Vienna, Austria; jobeiers@googlemail.com

**Keywords:** migraine aura, MRI, EEG, cortical spreading depolarization

## Abstract

Roughly one-third of migraine patients suffer from migraine with aura, characterized by transient focal neurological symptoms or signs such as visual disturbance, sensory abnormalities, speech problems, or paresis in association with the headache attack. Migraine with aura is associated with an increased risk for stroke, epilepsy, and with anxiety disorder. Diagnosis of migraine with aura sometimes requires exclusion of secondary causes if neurological deficits present for the first time or are atypical. It was the aim of this review to summarize EEG an MRI findings during migraine aura in the context of pathophysiological concepts. This is a narrative review based on a systematic literature search. During visual auras, EEG showed no consistent abnormalities related to aura, although transient focal slowing in occipital regions has been observed in quantitative studies. In contrast, in familial hemiplegic migraine (FHM) and migraine with brain stem aura, significant EEG abnormalities have been described consistently, including slowing over the affected hemisphere or bilaterally or suppression of EEG activity. Epileptiform potentials in FHM are most likely attributable to associated epilepsy. The initial perfusion change during migraine aura is probably a short lasting hyperperfusion. Subsequently, perfusion MRI has consistently demonstrated cerebral hypoperfusion usually not restricted to one vascular territory, sometimes associated with vasoconstriction of peripheral arteries, particularly in pediatric patients, and rebound hyperperfusion in later phases. An emerging potential MRI signature of migraine aura is the appearance of dilated veins in susceptibility-weighted imaging, which may point towards the cortical regions related to aura symptoms (“index vein”). Conclusions: Cortical spreading depression (CSD) cannot be directly visualized but there are probable consequences thereof that can be captured Non-invasive detection of CSD is probably very challenging in migraine. Future perspectives will be elaborated based on the studies summarized.

## 1. Introduction

Migraine with aura (MwA) comprises 20–30% of migraine sufferers [1]. In MwA, the aura usually precedes the headache attack, although headache is frequently perceived during the aura [2] and aura may begin after the headache phase has commenced [3]. Migraine auras may be followed by less distinct headaches or no headache. Most patients with MwA also have migraine without aura (MwoA) [3]. During the migraine aura, the patient experiences transient neurological deficits, such as visual, sensory, speech, motor, brainstem, or retinal symptoms, where visual disturbance is the most frequent symptom. Gradual spread of symptoms, duration of individual symptoms from 5–60 min, successive occurrence of symptoms, as well as positive and negative symptoms belong to the typical characteristics [2]. MwA seems more frequently associated with anxiety and with epilepsy than migraine without aura [4,5]. Further peculiarities of MwA include its roughly two-fold increased risk for vascular disease such as ischemic stroke [6] and its probable association with silent brain lesions [7]. Based on clinical observations of gradual spread of migraine aura symptoms corresponding to a velocity of approximately 3 mm per minute, it has been suggested that cortical spreading depression (CSD) is the pathophysiologic correlate of migraine aura [8]. The diagnosis of migraine is based on diagnostic criteria of the International Headache Society [3]. If presentation is atypical, neuroimaging and other diagnostic procedures may be necessary to rule out secondary headache.

Ischemic stroke is an important differential diagnosis for MwA, as it may have overlapping symptoms [9]. This is of particular interest when migraine aura present for the first time.

The present review will summarize EEG and MRI studies acquired during migraine aura, with a particular focus on clinically widely available modalities and integrate this into current pathophysiological concepts.

## 2. Methods

### Search Strategy

A literature search in the database Pubmed was conducted up to December 2022 using the search terms “migraine” AND “aura” AND “EEG” OR “electroencephalography”. In addition, we searched for “migraine” AND “aura” AND “neuroimaging” OR “MRI” OR “magnetic resonance imaging”. Relevant articles cited in other reviews were retrieved and references from original papers were screened. We included experimental studies, case series, and reports specifically investigating EEG and MRI alterations during aura or shortly thereafter. Articles were evaluated based on the description of migraine diagnosis.

## 3. Narrative Summary of studies

### 3.1. EEG Studies during Migraine Aura

As the duration of typical migraine aura is usually less than 1 h, ictal recordings during the aura phase are scarce. In contrast, duration of aura in hemiplegic migraine is longer and can last up to several days or even longer, which facilitates ictal recording. A summary of ictal EEG studies during aura is provided in Table 1. During visual auras, the EEG showed no abnormalities related to aura, although transient focal slowing in occipital regions has been historically observed during migraine aura [10,11,12]. One study which did not find specific abnormalities during migraine aura used a limited number of EEG electrodes [10]. A quantitative EEG study revealed contralateral decrease in α-power followed by an increase in bilateral frontal δ-power during visual aura in children [13]. During prolonged auras including motor symptoms or decreased levels of consciousness in some patients, occipitotemporal slowing evolving to occipital slowing over the affected hemisphere was detected, with correlation to cerebral hypoperfusion in SPECT [14] and decreased cerebral blood flow velocity in transcranial doppler sonography. Interestingly, EEGs performed within 3 h were normal in this study [14]. In this retrospective study, the blood flow abnormalities seemed to precede and persist throughout the period of slowing in EEG.

### 3.2. EEG Studies during Aura in Hemiplegic Migraine

In familial hemiplegic migraine (FHM) and migraine with brainstem aura, significant EEG abnormalities have been described, including slowing over the affected hemisphere or bilaterally or suppression of EEG activity [15,16,17]. It has to be kept in mind that FHM attacks can last several days and may be associated with an encephalopathic syndrome with fever, confusion, and altered levels of consciousness in some cases [18,19]. Epileptic seizures may occur in all FHM subtypes, in temporal relation to the FHM attack, independently from migraine attacks, or in childhood [19,20,21]. A systematic summary of EEG studies in FHM is given in Table 2

### 3.3. EEG during Migraine with brainstem aura

In children with migraine with brainstem aura, diffuse polymorphic high-voltage sub-δ to δ activity was recorded within 4 h of symptom onset [28], whereas, when symptoms fade or have resolved, slowing was confined to posterior regions. Reports of interictal occipital spikes and spike-wave activity in children with migraine with brain stem aura (formerly called “basilar migraine”) have already been critically discussed [28] as some of the early reports may be consistent with benign occipital lobe epilepsy, which has visuals symptoms and headache as cardinal symptoms.

### 3.4. MRI studies during Migraine Aura

Neuroimaging is indicated to rule out secondary causes if clinical presentation of migraine is not typical or if a new aura type occurs for the first time. Similar as with EEG-studies, ictal MRI acquisitions are scarce during migraine aura due to the short duration. Hence, only a few experimental studies investigated migraine aura in the early phases in a small number of subjects, while more patients with prolonged symptoms were reported in studies using data from emergency settings.

#### 3.4.1. Perfusion MRI Studies

During typical spontaneous migraine attacks with visual aura, a decrease in occipital perfusion was measured in the affected occipital lobe without clear restriction to vascular territories in four subjects [29]. Specifically, a decrease in cerebral blood flow (CBF), and to a lesser extent blood volume (CBV), and a corresponding increase in mean transit time of contrast agent (MTT) were observed. In one subject in whom repeated measurements were available during the aura, a spreading of hypoperfusion from posterior to anterior was noted, with mild hypoperfusion persisting into the headache phase without evidence of hyperperfusion in the headache phase. No diffusion abnormalities were observed during the migraine aura in any of the subjects [29]. Although this study investigated only a small number of participants, the beauty of this work is the acquisition of ictal MRIs in migraine with spontaneous typical visual aura, including a detailed clinical description. This study was extended to a total of seven cases during spontaneous visual aura, and patients with MwoA were also investigated. Moderate hypoperfusion during aura was confirmed in all subjects, with a mean 27% decrease in CBF, a 15% decrease in CBV, and a 32 % increase in MTT in the clinically affected occipital visual cortex. An illustrative example of perfusion changes shortly after migraine aura is given in Figure 1A. Later in the headache phase, a potential hyperperfusion was detected, corroborating previous PET studies [30]. In this study, hypoperfusion lasted into the headache phase and was followed by hyperperfusion. No perfusion abnormalities were observed during MwoA attacks [30]. In a retrospective clinical study in the emergency room setting, MRI hypoperfusion without restriction to vascular territories was observed in 70% of patients, similar to the pattern described above: increased MTT and time to peak (TTP), decreased CBF and minimal decrease in CBV [31]. In this study, where MRI was performed with a mean latency of 3 h, no hyperperfusion was observed. Diffusion-weighted imaging (DWI) and follow-up MRI imaging excluded cerebral ischemia [31]. In a similar retrospective study in the stroke setting including subjects with motor symptoms in 33%, perfusion abnormalities were seen in 54.4% of patients with migraine with aura, usually in more than one vascular territory [32]. In this study, in which perfusion ratios between affected and unaffected sides were calculated, CBF and CBV were more altered than TTP and MTT and were not limited to the territory of one artery in most cases. TTP was only moderately increased in MwA, in contrast to ischemic stroke [32]. MRA was unremarkable in about 80% of patients, while the remainder showed subtle abnormalities like decreased peripheral arterial MCA/PCA (medial/posterior cerebral artery) branch visualization or a dilatation of MCA or PCA. Diffusion studies were normal in all migraine subjects but one, which will be discussed below. Similarly, a patient with visual, sensory symptoms and aphatic symptoms lasting for 2 h showed widespread reduction in cerebral perfusion without alterations in DWI sequences in addition to venous enlargement over the affected hemisphere which will be discussed below [33]. Hypoperfusion was seen with arterial spin labeling (ASL) perfusion MRI in children in a stroke setting with migraine with aura, which was associated with vasospasm in small vessels affecting MCA and PCA in 58% of cases [34]. None of these patients had diffusion abnormalities. Another multimodal MRI study in pediatric patients in whom stroke was ruled out by negative DWI studies confirmed hypoperfusion in the majority of patients (8/12) with a transition to rebound hyperperfusion from 5.5. to 11 h. Decreased perfusion was associated with vasoconstriction in the MCA territory in most cases and also with decreased signal in SWI [35].

In the seminal paper by Hadjikani et al., a total of five migraine attacks with aura were studied in three male subjects with MwA with Fmri [36]. Two attacks were triggered in one subject by basketball training, while attacks occurred spontaneously in two subjects and were scanned 15–20 min after onset. During perception of the aura, a transient increase in mean cerebral blood flow was observed, where reactivity to visual stimulation was suppressed. Then, mean blood flow decreased while response to visual stimuli remained suppressed, followed by gradual recovery of mean CBF and stimulus response. Perfusion-weighted imaging showed decreased CBF and CBV and increased MTT in the occipital cortex, confirming previous studies. A retinotopic progression of blood flow disturbances over the visual cortex was observed consistent with migration of visual aura from central to peripheral visual fields. In summary, these blood flow characteristics were consistent with CSD in several aspects, such as initial short-lived cortical hyperemia, followed by more prolonged decrease in CBF, reduced stimulus-induced cortical responsiveness, and characteristic spread of CBF alterations over the affected visual cortex.

In contrast to hypoperfusion during migraine aura as detailed above, most studies on hemiplegic migraine show widespread cerebral hyperfusion during the phase with neurological deficits [17,37,38], which resolved in the interictal phase. As carefully elaborated by Hansen based on two cases and a critical review of the literature, hemiplegic migraine aura probably also begins with cerebral hypoperfusion, changing to hyperperfusion in later phases, as has been observed in MwA (“Classical migraine,” including hemiplegic cases [39]).

It shall be briefly mentioned that perfusion abnormalities in MwA have been described also in the headache and interictal phases. During headache, hyperperfusion was found in the brainstem and in occipital regions [40], whereas hyperperfusion in higher visual cortex areas was a feature of MwA in the interictal phase [41].

**Figure 1 diagnostics-13-02844-f001:**
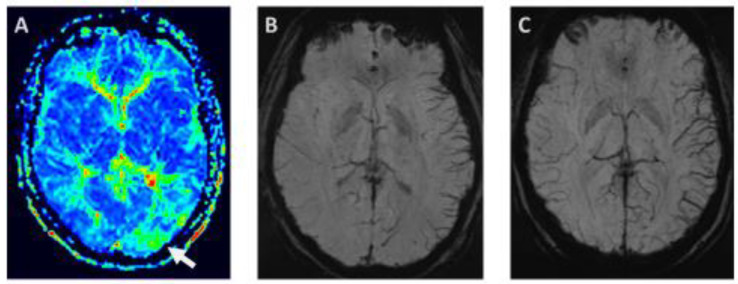
(**A**) Perfusion MRI shows an increase in mean transit time (MTT) in a 44-year-old female patient 200 min after the onset of an aura with visual and speech disturbance (abnormality indicated with white arrow). She had bifrontal headache while the aura had already remitted. (**B**) Susceptibility-weighted imaging (SWI) in a 34-year-old female demonstrates prominent cortical veins in the left hemisphere after a typical migraine aura with hemianopia, motor aphasia, and right-sided hemihypoaesthesia. During the scan, she still had the visual disturbance and dizziness together with hemicranial right-sided headache on right side. (**C**) Susceptibility-weighted imaging (SWI) in a 19-year-old female patient shows prominent focal veins after remission of a typical migraine aura consisting of right-sided paraesthesia and motor aphasia. During the scan, she had bifrontal headache. Images of the presented cases have not been published separately but are in line with recent studies [42].

#### 3.4.2. Susceptibility—Weighted Imaging—Prominent focal veins—Index Vein 

In a patient with MwA who had, after typical scintillating scotoma, developed headache, obscuring of peripheral branches of the contralateral posterior cerebral artery (PCA) was initially observed. About 45 min after aura onset, this was followed by normalization of peripheral arterial branches and enlargement of veins in susceptibility-weighted imaging (SWI). Later, in the headache phase (90 min after aura onset), dilation of the PCA and an increase in venous enlargement was described [43]. Such changes in SWI on the hemisphere affected by migraine aura have been confirmed in both adult and pediatric patients who were investigated in the emergency room setting for suspected stroke. Prominent focal veins without diffusion abnormalities have been described in 18.8% of adult patients during MwA attacks with a mean symptom duration of 5 h with inferior-posterior temporal or occipital predilection and have been related to neurovascular uncoupling during migraine aura [44]. In this study, about one-third had ongoing aura symptoms in the emergency room. The side of imaging abnormalities correlated with clinical symptom lateralization. In children, venous engorgement predominantly in occipital and temporal regions was found in 61.6 % of patients up to 9 h after symptom onset and showed good correlation with reduced perfusion, i.e., increased TTP [42]. The higher percentage of children with prominent focal veins was explained by a higher percentage of patients still experiencing aura symptoms during neuroimaging. Illustrative examples of venous dilatations in MwA are given in Figure 1B,C. Dilatation of contralateral sulcal veins in association with hypoperfusion, i.e., decreased CBF and increased MTT, has been described together with vasoconstriction in pediatric patients with hemiplegic migraine as early as 3 h after symptom onset [45]. Venous abnormalities are probably not specific for migraine and have been described also in ischemic stroke [46], whereas narrowed cortical veins and hyperperfusion have been reported in status epilepticus [47]. Our group has suggested clinical utility of a prominent sulcal vein in SWI, which could be termed as an “index vein” pointing towards the cortical area corresponding to the clinical presentation [48]. In a case series of six patients in an acute stroke setting a prominent focal vein, the “index vein”, draining the cortical area responsible for aura symptoms was described. The authors suggested more focal SWI abnormalities in migraine aura as opposed to stroke or epilepsy, which could be explained by gradual development of migraine symptoms. Indeed, such an “index vein” can almost exclusively be found in migraine aura with a prevalence of 17% in the acute setting compared to 2% in ischemic stroke, 4% in epileptic seizures, and 1% in acute neurological deficits of other origin. This sign thus has a very high specificity when present and could be useful in the clinical setting of acute neurological deficits [49].

#### 3.4.3. Diffusion Abnormalities

In MwA with atypical presentation, DWI is used to rule out migrainous infarction. There are a few reports of reversible alterations of DWI during prolonged migraine aura not associated with brain damage in follow-up MRI. Reversible lesions in the splenium of the corpus callosum with increased signal on DWI and T2/FLAIR have been described in migraine with aura with sensory–motor symptoms lasting for 48 h [50] and in status migrainosus with intermittent focal symptoms [51]. Similar lesions are now termed cytotoxic lesions of the corpus callosum (CLOCCS) and were formerly known as meningitis/encephalitis with reversible splenial lesions (MERS) or as reversible splenial lesion syndrome (RESLES). This imaging phenotype has been related to a variety of etiologies, including epilepsy, metabolic disturbances, or demyelinating disease [50]. 

In one multimodal MRI-study (see perfusion studies), one subject with complex congenital heart disease showed a small diffusion abnormality in the left caudate nucleus, along with perfusion restriction in the left parietal and occipital lobes. Diffusion abnormality was considered as an incidental finding related to cardiac embolism and not to migraine aura in this subject [32]. In our opinion, however, it should be kept in mind that embolism may trigger CSD and thus this patient might have had secondary migraine aura-like symptoms [52]. In a patient with prolonged aura with hemianopia, hemineglect, paresthesia, and confusion, MRI showed related alterations in FLAIR and DWI in the cortex 12 h after onset, which were reversible on follow-up [53]. 

Hemiplegic migraine may be associated with cerebral edema readily seen on T2-weighted imaging including FLAIR [54]. Diffusion restriction in the cortex and in the deep grey matter has been described in FHM and sporadic hemiplegic migraine [18,55,56], but this does not seem to be the rule [17,38]. In hemiplegic migraine, contrast enhancement has been occasionally reported in addition to cortical edema [18,57].

During a migraine attack with prolonged visual disturbance and bilateral paresthesia for 4 days, occipital alteration in apparent water diffusion coefficient (ADC) without DWI abnormalities have been reported, which were reversible on follow-up [58]. Similarly, fully reversible DWI lesions spreading from occipital to temporal lobes have been described in a subject with persistent visual aura without infarction [59].

## 4. Discussion

It is widely accepted that CSD is the pathophysiological correlate of migraine aura, but the direct proof is still missing. The link between CSD and migraine headache is the activation of the trigeminovascular system, directly or via activation of meningeal nociceptors [60,61,62,63]. The studies presented show electrophysiological and blood flow signatures of CSD, which will be summarized below, without direct demonstration thereof. With a few exceptions, prolonged auras with several symptoms were studied. Thus, it remains unclear which of the findings would also be present during typical migraine aura. In the present review, familial hemiplegic migraine (FHM) was included, assuming a similar pathophysiology, although this could be challenged.

### 4.1. EEG Signatures of the Migraine Aura

EEG during short-lived auras is probably normal and might show α-power reduction and increase in δ-activity in quantitative studies in children, although evidence is limited, based on few studies. In contrast, in FHM and migraine with brainstem aura, significant EEG abnormalities were found, including slowing over the affected hemisphere, bilaterally, or suppression of EEG activity [15,16,17] and epileptiform potentials.

#### Familial Hemiplegic Migraine, Epilepsy and EEG

All types of FHM have been associated with epilepsy, where seizures may occur in temporal relation to the attack, without relation to the migraine attack or during childhood. Patients with FHM attacks showing epileptic potentials in addition to slowing, without convulsive seizures were occasionally treated with anti-seizure medication such as phenytoin or levetiracetam [25]. Transgenic animal models have demonstrated increased susceptibility for CSD as well as for seizures in FHM mice as reviewed elsewhere [64]. FHM Type 1 is caused by mutations in the CACNA1A (calcium voltage-gated channel subunit alpha1 A –gene; calcium gene coding for the voltage-dependent P/Q type Ca channel) and is frequently associated with cerebellar atrophy, ataxia, and nystagmus [19,65]. Mutations in the CACNA1A gene were found in episodic ataxia type 2 and spinocerebellar ataxia type 6 and epilepsy with genetic etiology [65]. FHM Type 2 is caused by mutations in the ATP1A2 gene coding for Na/K-ATPase [20,66]. FHM3 is caused by mutations in the SCN1A gene (sodium voltage-gated channel alpha subunit 1) coding for a sodium channel and has been related to epilepsy syndromes such as Dravet syndrome and generalized epilepsy with febrile seizures plus [21]. Considering these relations to epilepsy, the occurrence of epileptic potentials in FHM are probably to be considered as true epileptic. This supports the clinical relevance of EEG in FHM attacks, even in absence of clinical seizures to diagnose non-convulsive status epilepticus, which necessitates antiepileptic treatment. 

### 4.2. MRI Signatures of Migraine Aura 

CSD is associated with a short-lived increase in cerebral perfusion lasting for 1–2 min according to animal experiments, followed by a more prolonged decrease in CBF [67], although hemodynamic response to CSD may vary according to the animal preparation [64]. This CBF signature of CSD was shown in triggered MwA attacks using functional MRI [36] and in historical studies using intracarotid labeled xenon injection [68], supporting the notion of a brief increase in CBF followed by a longer period of hypoperfusion (spreading oligemia) that usually involves more than one vascular territory. Thereafter, hyperperfusion was reported, but the time relation to headache appearance seems less clear [69]. A schematic summary of blood flow changes and other imaging features of MwA attacks is given in Figure 2 to visualize the temporal appearance of MRI alterations. Note that diffusion restrictions, contrast enhancement, or flair hyperintensity have only been found in long-lasting complex auras as in FHM. The transition from hypoperfusion to rebound hyperperfusion is estimated to occur after about 5–6 h, with variability between studies. Of note, these findings derive from case series done in the emergency setting using clinical routine perfusion studies. Serial measurements in the same subjects are available only in hemiplegic migraine. A more recent study reported hyperperfusion in visual areas and in the brainstem bilaterally in the headache phase of typical MwA attacks [40].

In pediatric patients, hypoperfusion was frequently associated with decreased diameters of distal arteries [34,35], whereas this association was less evident in adults [32]. Another imaging hallmark of migraine aura is signal attenuation in SWI consistent with prominent focal veins or venous engorgement, probably reflecting increased oxygen demand during CSD [43]. The “index vein” points to the cortical region correlating with migraine symptoms and helps to differentiate migraine-aura-related SWI abnormalities from SWI changes in stroke or epileptic seizures.

Several features of CSD during migraine aura have been identified in the seminal paper by Hadjikhani [36], including initial hyperemia followed by spreading oligemia and reduced responsivity of the cortex to visual stimuli. A magnetoencephalographic study during migraine auras was consistent with a direct current shift [70]. Further evidence for CSD being the correlate for migraine aura comes from studies on evoked potentials. During visual aura, a suppression of visual evoked potentials was found [71], consistent with reduced cortical function and reactivity. During an attack of FHM, motor- and somatosensory-evoked potentials over the affected hemisphere were suppressed [17].

## 5. Conclusions, Context, and Future Perspectives

Features of CSD were captured with various MRI modalities in animals [72], suggesting possible future translational approaches to human studies. As signatures of CSD, such as slow potential shifts, and suppression of faster activity have been defined in surface EEG in patients with severe neurological disease in the intensive care setting [73], it is an interesting question if this would also be true for migraine aura. In epilepsy patients on epilepsy monitoring units, hints for CSD have been described in the surface EEG [74]. However, the detection of direct current shifts related to CSD over the intact scalp may be methodologically very challenging. Defining which of the imaging modalities are most appropriate for assessing migraine aura deserves further study in longitudinal designs. 

Neurophysiological studies with event-related potentials and neuroimaging studies have been developed in machine learning settings for automated classification of ictal and interictal states in migraine or patients from healthy controls [75,76,77]. An automated classification algorithm based on cortical features, such as cortical thickness, surface area, volume, or folding index from MRI post processing, could distinguish not only between MwA patients and healthy controls but also between migraine patients with simple and complex aura with high accuracy [76]. Interestingly, migraine with complex aura (visual somatosensory, dysphasic symptoms) has been related to impaired cognitive processing in an event-related potential study using the oddball paradigm evidenced by significant differences in the P3 component [78] and distinct structural alterations in MRI after cortical surface-based morphometric analyses [79]. Thus, the complexity of migraine aura should be considered in future neurophysiological and neuroimaging studies. MwA patients showed evidence for altered semantic processing in evoked potential studies in the migraine-free state [80], which could be analyzed across the migraine cycle in MwA. 

The typical short-lived aura in humans is of particular interest. One option to cover the initial phase of auras would be to trigger auras. Amongst various triggers, such as normobaric hypoxia, nitroglycerin or CGRP, the ATP-sensitive potassium-channel (K^+^-ATP) opener levcromakalim seems to be reliable [81]. For the other option, registering spontaneous auras by accident, i.e., while subjects are in the scanner or connected to EEG, a substantial amount of luck would be necessary, as would be an alertness from the medical team to notice the preciousness of such moment, i.e., to be ready to catch the aura if one can.

## Figures and Tables

**Figure 2 diagnostics-13-02844-f002:**
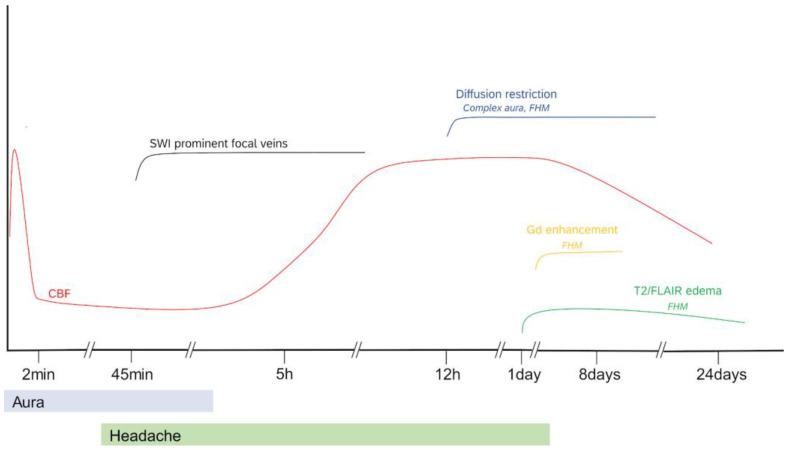
Time course of blood flow characteristics and MRI abnormalities in migraine aura, prolonged and hemiplegic migraine attacks. Note that timing may vary between subjects; not every subject develops all features of the schematic description. In pediatric patients, hypoperfusion was frequently associated with vasoconstriction in MRA. Modified from [69].

**Table 1 diagnostics-13-02844-t001:** EEG findings during aura in MwA.

Ref.	Diagnosis	Relevant Findings	Patients Studied during Aura	Age Range
Dow et al., Lancet 1947 [11]	Migraine, subgroup with visual aura	No focal changes during aura	6	14–56 Not specified in subgroup with MwA
Lauritzen et al., Cephalalgia 1981 [10]	Classical migraine	No changes or unspecific changes during aura as well as interictally	3	16, 32
Seri et al., Cephalalgia 1993 [13]	Classical migraine	Contralateral decrease in α-power, followed increase in bilateral frontal δ-power; in the headache phase, increase in δ-power in posterior temporal and occipital regions	10	8–14
Parain et al., Cephalalgia 2007 [14]	Prolonged aura 4–24 h with or without weakness	Slowing in recordings after 3 h in temporo-occipital regions or diffuse, occiptal slowing on the second day	11	8–15

**Table 2 diagnostics-13-02844-t002:** EEG Studies during Hemiplegic Migraine.

Ref.	Diagnosis	Main Findings	Patients Studied during Aura	Age Range
Chan et al. J Clin Neurosci 2008 [22]	FHM Type 1 CANA1A mutation, 218	Depressed EEG activity with low amplitudes over the hemisphere contralateral to hemiparesis, followed by high amplitude δ-activity; 1 patient who also suffered from childhood epilepsy showed in addition paroxysmal activity in the ϑ–δ range over the hemisphere ipsilateral to the hemiparesis	3	12–19
Fitzsimons et al. Brain 1985 [23] Kors et al. Ann Neurol 2001 [24]	FHM Type 1 CACNA1A, S218L	Slow waves over hemisphere related to the paresis and “paroxysms” over the contralateral hemisphere	4	20–41
Murphy J Clin Neurophysiol 2018 [18]	FHM Type 2, mutations in the ATP1A2 gene	Excess of slow-wave-activity in the ϑ–δ range; 1 case had encephalopathic features in the EEG with periods of suppression, which was still asymmetric. This case had cerebral edema of the affected hemisphere with signal increase in FLAIR sequences, diffusion restriction in grey matter and contrast enhancement of ipsilateral mesial temporal lobe	5	12–32
Schwarz et al. J Neurol Sci 2018 [25]	Sporadic hemiplegic migraine, mutations in the SCN1A-gene	Periodic interictal epileptic discharges over the hemisphere related to paresis in addition to continuous slow-wave abnormalities. This EEG pattern probably refers to lateralized periodic discharges (LPD) [26]. This patient was treated with levetiracetam and phenytoin and had cortical signal alterations on T2-weighted MRI but no diffusion restriction	1	14
Chastan et al. Neurophysiol Clin 2016 [27]	Sporadic hemiplegic migraine, mutations in the SCN1A-gene	Posterior slow waves were observed as soon as 15 min after onset of scintillating scotoma with spreading to anterior regions in parallel to symptom evolution to hemiparesis	1	19

## Data Availability

Not applicable.

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
