# Peer review of "Migraine Aura—Catch Me If You Can with EEG and MRI—A Narrative Review"

_diagnostics, 2023, doi:10.3390/diagnostics13172844_

Round 1
Reviewer 1 Report (Previous Reviewer 2)
Writing quality is poor. There are a lot of errors such as inadequate introduction of abbreviations and inadequate and unsystematic Figure referencing. The structure of the paper chapters is confusing. "Chapter 3. - Narrative Summary of studies" and "Chapter 4. - Discussion" are quite similar.
Figure 1 is referenced twice in the same chapter:
1. “An example of perfusion changes 150 shortly after migraine aura is given in Fig. 1.” and “An illustrative image of perfusion alteration in MwA is provided in Figure 1A”
while it is not clear neither from the text nor from the image caption whether the images shown present results from a published study or not.
Figure 3. would be more adequate for a textbook than a science paper and should be removed.
There is no summary of the results and conclusions which may be relevant for better understanding of migraine pathophysiology. Moreover, there are multitude of MRI and EEG studies focused specifically on migraine with aura conducted in the interictal period. This review would only make sense if those studies are contrasted and compared.
Author Response
Writing quality is poor. There are a lot of errors such as inadequate introduction of abbreviations and inadequate and unsystematic Figure referencing. The structure of the paper chapters is confusing. "Chapter 3. - Narrative Summary of studies" and "Chapter 4. - Discussion" are quite similar.
A: All abbreviations were carefully reviewed and supplemented, where necessary. For instance the abbreviations for MCA/PCA (medial/posterior cerebral artery) were provided. To improve clarity and readability, gene names were explained in the discussion. A few typos such as e.g., FMH instead of FHM were corrected. Figure referencing was revised. The discussion provides main findings
Figure 1 is referenced twice in the same chapter:
- “An example of perfusion changes 150 shortly after migraine aura is given in Fig. 1.” and “An illustrative image of perfusion alteration in MwA is provided in Figure 1A”
while it is not clear neither from the text nor from the image caption whether the images shown present results from a published study or not.
A: Thank you for his comment. The double referencing in the text was removed. The following statement was added to the caption of Fig 1: Images of the presented cases have not been published separately but are in line with recent studies [1].
Figure 3. would be more adequate for a textbook than a science paper and should be removed.
A: Figure 3 was removed.
There is no summary of the results and conclusions which may be relevant for better understanding of migraine pathophysiology. Moreover, there are multitude of MRI and EEG studies focused specifically on migraine with aura conducted in the interictal period. This review would only make sense if those studies are contrasted and compared.
A: Key findings of the reviewed articles are summarized in the discussion and discussed in the context of cortical spreading depression. While previous review articles such as for example [2] [3] compared neuroimaging studies during aura and ictal states, it was the aim of the present work to specifically shed light on findings during the aura. A particular focus was put on the setting of included studies (e.g. inclusion of patients from stroke or emergency care settings vs. patients with previously known migraine who could be studied with dedicated research protocols). The temporal appearance and evolution of neuroimaging findings during aura were highlighted. For example cortical edema or diffusion alterations were only described in patients with prolonged auras. Considering the multitude of studies, interictal studies were only marginally considered where relevant.
References:
- Kellner-Weldon, F.; Lehmann, V.F.; Breiding, P.S.; Grunder, L.; Muri, R.; Pastore-Wapp, M.; Bigi, S.; Wiest, R.; El-Koussy, M.; Slavova, N. Findings in susceptibility weighted imaging in pediatric patients with migraine with aura. Eur J Paediatr Neurol 2020, 28, 221-227, doi:10.1016/j.ejpn.2020.05.008.
- Arca, K.N.; VanderPluym, J.H.; Halker Singh, R.B. Narrative review of neuroimaging in migraine with aura. Headache 2021, 61, 1324-1333, doi:10.1111/head.14191.
- Karsan, N.; Silva, E.; Goadsby, P.J. Evaluating migraine with typical aura with neuroimaging. Front Hum Neurosci 2023, 17, 1112790, doi:10.3389/fnhum.2023.1112790.

Reviewer 2 Report (Previous Reviewer 1)
The manuscript is much improved.
There are sections that needs to be edited so the paragraphs will be of a reasonable length (3.4.1. Perfusion-MRI Studies, 3.4.2. Susceptibility-Weighted Imaging - Prominent focal veins – Index Vein).
Author Response
- Kellner-Weldon, F.; Lehmann, V.F.; Breiding, P.S.; Grunder, L.; Muri, R.; Pastore-Wapp, M.; Bigi, S.; Wiest, R.; El-Koussy, M.; Slavova, N. Findings in susceptibility weighted imaging in pediatric patients with migraine with aura. Eur J Paediatr Neurol 2020, 28, 221-227, doi:10.1016/j.ejpn.2020.05.008.
- Arca, K.N.; VanderPluym, J.H.; Halker Singh, R.B. Narrative review of neuroimaging in migraine with aura. Headache 2021, 61, 1324-1333, doi:10.1111/head.14191.
- Karsan, N.; Silva, E.; Goadsby, P.J. Evaluating migraine with typical aura with neuroimaging. Front Hum Neurosci 2023, 17, 1112790, doi:10.3389/fnhum.2023.1112790.
Response to reviewer 2
There are sections that needs to be edited so the paragraphs will be of a reasonable length (3.4.1. Perfusion-MRI Studies, 3.4.2. Susceptibility-Weighted Imaging - Prominent focal veins – Index Vein).
A: The sections indexed above were slightly shortened. Details on magnitude of CBF changes and automated quantitative evaluation of SWI-alterations were considered as not essential and could be reomoved. However, we considered a detailed description of the study setting as essential for study interpretation.
Reviewer 3 Report (New Reviewer)
In this paper, the authors provide a narrative review of EEG and MRI findings during migraine aura, with a specific emphasis on identifying distinctive EEG and MRI patterns. They also discuss the integration of these findings into current pathophysiological concepts, along with summarizing the diagnostic implications from the existing literature.
The subject matter is both timely and clinically relevant. The paper is adeptly written and well-organized, effectively presenting the outcomes of EEG and MRI investigations in a lucid manner. The practical value of the article for clinicians is highlighted, particularly in summarizing the results of relevant studies.
However, in my opinion there is room for improvement in the discussion and conclusion sections.
- To enhance the impact of the paper, it would be advantageous to offer a more comprehensive discussion of the identified EEG and MRI signatures, along with broader insights into the implications stemming from the reported findings.
- In addition, the study was clearly focused on blood flow signatures assessed with MRI. In my opinion the readers would benefit of short discussion regarding the agreement of findings of reported MRI findings with findings on this topic obtained with other neuroimaging techniques, primarily with CT Perfusion (such as: 10.1007/s10072-020-04476-5, 10.1177/0333102414523339).
Addressing the mentioned areas for improvement would further enhance the paper's impact.
Author Response
A: We thank the Academic Editor for his comments appreciating the revised article.
The tile was changed as follows: “Migraine aura – Catch me if you can with EEG and MRI – A narrative review”
The purpose of the review was now stated in the abstract: “It was the aim of this review to summarize EEG and MRI findings during migraine aura in the context of pathophysiological concepts”.
Round 2
Reviewer 1 Report (Previous Reviewer 2)
Even though this work is focused on reviewing the EEG and MRI studies during migraine aura, there are other important EEG studies exploring different responses (ERPs) that may reflect specific features of migraine aura and its complexity, and these should be mentioned in the discussion. Moreover, recent MRI and EEG studies explore specific functional and structural signatures of migraine aura, which can have a potential of better classification and diagnosis of migraine:
Zhu, B., Coppola, G. and Shoaran, M., 2019. Migraine classification using somatosensory evoked potentials. Cephalalgia, 39(9), pp.1143-1155.
Petrusic, I., Jovanovic, V., Kovic, V. and Savic, A.M., 2022. P3 latency as a biomarker for the complexity of migraine with aura: Event-related potential study. Cephalalgia, 42(10), pp.1022-1030.
Petrusic, I., Jovanovic, V., Kovic, V. and Savic, A., 2021. Characteristics of N400 component elicited in patients who have migraine with aura. The journal of headache and pain, 22(1), pp.1-9.
Mitrović, K., Petrušić, I., Radojičić, A., Daković, M. and Savić, A., 2023. Migraine with aura detection and subtype classification using machine learning algorithms and morphometric magnetic resonance imaging data. Frontiers in Neurology, 14.
Hsiao, F.J., Chen, W.T., Wang, Y.F., Chen, S.P., Lai, K.L., Coppola, G. and Wang, S.J., 2023. Identification of patients with chronic migraine by using sensory-evoked oscillations from the electroencephalogram classifier. Cephalalgia, 43(5), p.03331024231176074.
Abagnale, C., Di Renzo, A., Sebastianelli, G., Casillo, F., Tinelli, E., Giuliani, G., Tullo, M.G., Serrao, M., Parisi, V., Fiorelli, M. and Caramia, F., 2023. Whole brain surface-based morphometry and tract-based spatial statistics in migraine with aura patients: difference between pure visual and complex auras. Frontiers in Human Neuroscience, 17, p.169.
These recent studies and methods, and relation to the body of knowledge about migraine aura during the attack should be discussed.
Author Response
Response to review:
Even though this work is focused on reviewing the EEG and MRI studies during migraine aura, there are other important EEG studies exploring different responses (ERPs) that may reflect specific features of migraine aura and its complexity, and these should be mentioned in the discussion. Moreover, recent MRI and EEG studies explore specific functional and structural signatures of migraine aura, which can have a potential of better classification and diagnosis of migraine:
Zhu, B., Coppola, G. and Shoaran, M., 2019. Migraine classification using somatosensory evoked potentials. Cephalalgia, 39(9), pp.1143-1155.
Petrusic, I., Jovanovic, V., Kovic, V. and Savic, A.M., 2022. P3 latency as a biomarker for the complexity of migraine with aura: Event-related potential study. Cephalalgia, 42(10), pp.1022-1030.
Petrusic, I., Jovanovic, V., Kovic, V. and Savic, A., 2021. Characteristics of N400 component elicited in patients who have migraine with aura. The journal of headache and pain, 22(1), pp.1-9.
Mitrović, K., Petrušić, I., Radojičić, A., Daković, M. and Savić, A., 2023. Migraine with aura detection and subtype classification using machine learning algorithms and morphometric magnetic resonance imaging data. Frontiers in Neurology, 14.
Hsiao, F.J., Chen, W.T., Wang, Y.F., Chen, S.P., Lai, K.L., Coppola, G. and Wang, S.J., 2023. Identification of patients with chronic migraine by using sensory-evoked oscillations from the electroencephalogram classifier. Cephalalgia, 43(5), p.03331024231176074.
Abagnale, C., Di Renzo, A., Sebastianelli, G., Casillo, F., Tinelli, E., Giuliani, G., Tullo, M.G., Serrao, M., Parisi, V., Fiorelli, M. and Caramia, F., 2023. Whole brain surface-based morphometry and tract-based spatial statistics in migraine with aura patients: difference between pure visual and complex auras. Frontiers in Human Neuroscience, 17, p.169.
These recent studies and methods, and relation to the body of knowledge about migraine aura during the attack should be discussed.
A: We thank the reviewer for this comment. We introduced a paragraph in the discussion under the section now termed:
- Conclusions, Context and Future Perspectives
Discussing these papers in the context.
Neurophysiological studies with event-related potentials and neuroimaging studies have been developed in machine learning settings for automated classification of ictal and interictal states in migraine or patients from healthy controls [1-3]. An automated classification algorithm based on cortical features such as cortical thickness, surface area, volume or folding index from MRI post processing could distinguish not only between MwA patients and healthy controls but also between migraine patients with simple and complex aura at high accuracy [2]. Interestingly, migraine with complex aura (visual somatosensory, dysphasic symptoms) has been related to impaired cognitive processing in an evented-related potential study using the oddball paradigm evidenced by significant differences in the P3 component [4] and distinct structural alterations in MRI after cortical surface-based morphometric analyses[5]. Thus, the complexity of migraine aura should be considered in future neurophysiological and neuroimaging studies. MwA patients showed evidence for altered semantic processing in evoked potential studies in the migraine free state [6], which could be analyzed across the migraine cycle in MwA.
- Zhu, B.; Coppola, G.; Shoaran, M. Migraine classification using somatosensory evoked potentials. Cephalalgia 2019, 39, 1143-1155, doi:10.1177/0333102419839975.
- Mitrovic, K.; Petrusic, I.; Radojicic, A.; Dakovic, M.; Savic, A. Migraine with aura detection and subtype classification using machine learning algorithms and morphometric magnetic resonance imaging data. Front Neurol 2023, 14, 1106612, doi:10.3389/fneur.2023.1106612.
- Hsiao, F.J.; Chen, W.T.; Wang, Y.F.; Chen, S.P.; Lai, K.L.; Coppola, G.; Wang, S.J. Identification of patients with chronic migraine by using sensory-evoked oscillations from the electroencephalogram classifier. Cephalalgia 2023, 43, 3331024231176074, doi:10.1177/03331024231176074.
- Petrusic, I.; Jovanovic, V.; Kovic, V.; Savic, A.M. P3 latency as a biomarker for the complexity of migraine with aura: Event-related potential study. Cephalalgia 2022, 42, 1022-1030, doi:10.1177/03331024221090204.
- Abagnale, C.; Di Renzo, A.; Sebastianelli, G.; Casillo, F.; Tinelli, E.; Giuliani, G.; Tullo, M.G.; Serrao, M.; Parisi, V.; Fiorelli, M.; et al. Whole brain surface-based morphometry and tract-based spatial statistics in migraine with aura patients: difference between pure visual and complex auras. Front Hum Neurosci 2023, 17, 1146302, doi:10.3389/fnhum.2023.1146302.
- Petrusic, I.; Jovanovic, V.; Kovic, V.; Savic, A. Characteristics of N400 component elicited in patients who have migraine with aura. J Headache Pain 2021, 22, 157, doi:10.1186/s10194-021-01375-8.
Round 3
Reviewer 1 Report (Previous Reviewer 2)
Authors have successfully integrated changes according to my suggestions.
This manuscript is a resubmission of an earlier submission. The following is a list of the peer review reports and author responses from that submission.
Round 1
Reviewer 1 Report
The authors reviewed the literature on the EEG and MRI features of aura in migraine patients. This is a decent review and may be useful especially for residents and fellows, as it explains the different findings in each modality and the physiological background.
a. Major issues
1. The introduction and the discussion are very similar. It is unclear what the added value of each one is.
2. Section 2.2 (Cortical Spreading Depression – Pathophysiology of Migraine Aura) is not well integrated. This review focuses on the EEG and MRI features, thus, the authors should add to this section supporting the data from EEG and MRI studies.
b. Minor issues
1. Please explain the abbreviation – FMH-mutations when first used in the introduction.
2. Sections numbering is unclear. Is 2.1 a subsection of the introduction? If so, where is section 1? If not, then there should be a section 2 in bold (similar to 1. Introduction and 3. Discussion) followed by further subsections.
3. basilar migraine – this term is no longer in use, please use terms as they appear in the most recent ICHD classification.
4. Redundant title - Perfusion-MRI Studies
5. Figure 1B – please clearly mark the abnormality with an arrow or a circle.
6. Figure 2 – "Signals are filtered typically in the range between 0.5Hz and 40Hz" – this is incorrect. In clinical use, EEG is usually filtered between 0.5 and 70Hz in scalp EEG and is not filtered at all in ECoG.
7. table 1 should be after the EEG studies subsection.
8. Consider adding a figure with an EEG example of focal slowing during aura from an actual patient.
1. Spelling mistakes
- familiar hemiplegic migraine (FHM) – should be familial – correct in the abstract and in section 2.1.1
- over hemisphere realted to the paresis
- observed as soon as 15 minutes after onset off scintillating scotoma – should be of
- ECG electrode (figure 2)– should be ECoG
- CSD an the Link to Migraine Headache – should be and
- In contrast, in FMH and migraine with brain stem aur,a - should be aura,
- a longer period pf hypoperfusion - should be of
2. Please explain the abbreviation – FMH-mutations when first used in the introduction
3. Grammar – "were reported during an attack of FMH with hemiparesis and coma[27], who was diagnosed as FMH Type 1 CACNA1A, S218L mutation later" – should add a subject that "who" is related to
Reviewer 2 Report
The topic of the paper is interesting and relevant however the results are not presented adequately and in a manner required for a scientific paper.
Generally, this paper looks more like a chapter from a textbook on migraine than a review scientific paper. Metodology for acquiring relevant literature is not described. The title is misleading since the EEG studies in the ictal period are less systematically explored and described. Introduction of a table with EEG studies in the ictal period would be necessary.
Figures are not adequate. Figure 1 is taken from the authors own work which is not previously published and this is not a common practice in scientific review articles.
Figure 2. would be more adequate for a texbook than a science paper.
The chapters are written in a very similar manner, meaning that the discussion is written in a similar manner as previous chapters. There is no real summary of the results and conclusions which are relevant points of a review paper.
Writting quality is poor and not adequate for a research paper.
Minor:
P2. “Examples of own patients we could catch during or shortly after the aura has stopped are provided in Fig-ure 1 for illustrative purpose”
Please rephrase
P3. “One study with negative ictal findings used a limited number of EEG electrodes[14].”
What are negative ictal findings? Please clarify.
P3. “Interestingly, EEGs performed within 3 h were normal in this study[18].”
Check grammar